# Reduced health services at under-electrified primary healthcare facilities: Evidence from India

**Vivek Shastry** [1]*, **Varun Rai**[1,2]

**1** LBJ School of Public Affairs, University of Texas at Austin, Austin, Texas, United States of America,
**2** Mechanical Engineering Department, University of Texas at Austin, Austin, Texas, United States of America

* svivekshastry@utexas.edu

**Data Availability Statement:** All relevant data are within the paper and its Supporting Information files.

**Funding:** The authors received no specific funding for this work.

## Abstract

Primary healthcare systems worldwide suffer from major gaps in infrastructure and human resources. One key infrastructure gap is access to reliable electricity, absence of which can significantly affect the quantity and quality of healthcare services being delivered at rural primary health facilities. However, absence of granular empirical evidence is a barrier for quantitatively understanding the significance of electricity access as one of the determinants of access to reliable primary healthcare. Using data from India's District Level Household and Facility Survey, we develop zero-inflated negative binomial models with co-variates and state-level fixed effects to estimate the relationship between levels of electricity access and the quantity of basic health services delivered at Primary Health Centers (PHCs). We find that lack of electricity access is associated with a significant and large decrease in the number of deliveries (64 percent), number of in-patients (39 percent), and number of out-patients (38 percent). We further find that lower level of electricity access at primary health centers is disproportionately associated with adverse effects on women's access to safe and quality healthcare.

## 1. Introduction

United Nations SDG 3 on Health aspires to "*achieve universal health coverage, including. . . access to quality essential health-care services. . .for all*" by 2030, while SDG 7 on Energy aspires to "*ensure universal access to affordable, reliable and modern energy services*". Achieving SDG 7 may even be fundamentally tied to achieving SDG 3: owing to its potential impacts on medical services, health and safety, disease prevention and treatment, staff recruitment and retention, and administration and logistics, reliable electricity availability is understood to be an enabler of access to quality healthcare [1–3]. The focus of this paper is to explore that dependency empirically in the context of primary healthcare in India.

Rural health centers in developing countries suffer from significant constraints with accessing reliable electricity. A 2013 multi-country study estimated that only a third of the surveyed hospitals in sub-Saharan Africa had access to reliable electricity [4]. In another recent study,

**Competing interests:** The authors have declared that no competing interests exist.

half of the primary health centers in India reported not having reliable electricity access [5]. Beyond these aggregate findings, however, evidence on the variation in healthcare delivery in relation to electricity access is mostly anecdotal. A WHO report noted that a systematic literature review "*did not identify a single study in which linking energy access and health outcomes was the primary objective*" [1]. Absence of empirical evidence is a barrier for understanding the relative significance of electricity access as one of the determinants of access to reliable primary healthcare. SDG7's goal of ensuring access to modern energy services to all includes access to electricity as well as clean cooking services. Considerable research exists on the linkages between energy access and health outcomes at a household level, particularly on the impact of clean cooking [6–8]. The WHO report and our analysis in this paper specifically address the gap in literature on the linkages between energy access, particularly electricity access, and health outcomes at *health facilities*.

Recent literature has focused on providing better descriptive understanding. An analysis of health facilities in Senegal showed that half of the facilities did not have access to electricity [9]. A number of these facilities had medical equipment but no electricity to power them. A qualitative study of Malawian health facilities showed poor electricity access being associated with irregular water supply, poor sterilization, and poor working conditions [10, 11]. Another recent report provides a descriptive study of the availability of electricity at Primary Health Centers (PHCs) in India [12]. Their preliminary findings show that a higher proportion of PHCs which have regular electricity access are able to provide different health services, compared to health centers without electricity access. The authors also show that higher proportion of PHCs with regular electricity access had resident medical staff and critical medical equipment compared to PHCs without electricity access. We use these preliminary findings as a starting point for our analysis.

The unique contributions of this paper are: (1) drawing additional insights from descriptive analysis and (2) developing an empirical model to quantitatively establish the linkages between reliable electricity access and the delivery of primary healthcare. We study these linkages by focusing on PHCs in India, which are the main points of comprehensive primary care in the Indian public health system. India has a vast network of over 25,000 PHCs, which serve as the first point of access for mother and childcare, immunization, in-patient and out-patient care, emergency care, laboratory services, and sometimes basic surgical procedures. Reliable access to electricity may be expected to play a significant role in the ability of PHCs to adequately perform these functions. To put this in context, as of 2019 there were still 795 PHCs (about 5 percent of the total PHCs) without any electricity supply [13]. These PHCs without electricity supply cumulatively served at least 24 million rural individuals, assuming each PHC served 30,000 individuals on average. In fact, the proportion of PHCs and sub-centers without electricity has not changed substantially since 2015 (see Fig A1 in S1 Appendix).

The analysis presented in this paper provides robust empirical evidence to the hypothesis that lower levels of electricity access is associated with lower provision of health services and with lower availability of medical staff and functional equipment at rural primary health centers. We further find that lower level of electricity access at PHCs is disproportionately associated with adverse effects on women's access to safe and quality healthcare. These findings are relevant not only to the last-mile public health infrastructure in India, but also to many other parts of the developing world where unreliable electricity at public health facilities affects hundreds of millions of poor and underserved communities. The rest of this paper is organized as follows. We describe the data and the criteria for model specification in section 2. In section 3, we present the descriptive analysis and inferences from the empirical models. We conclude with a discussion in section 4.

## 2. Data and methods

### 2.1. Data source and preparation

We use data from the fourth round of India's District Level Household and Facility Survey [14] conducted in 2012–13. DLHS-4 contains data from a nationally representative sample of 8540 Primary Health Centers (PHCs) from across India, from all states except Gujarat and Jammu and Kashmir. These data include information on PHC characteristics, condition, infrastructure, staff, and the health services delivered. The final sample contains 7805 PHCs, which accounts for 32 percent of the total PHCs at the time of the survey. As of March 2012, there were a total of 24049 PHCs in India. As of March 2018, the total number of PHCs was only slightly higher at 25743 [15]. We also use pooled data from DLHS-3 and DLHS-4 to test the robustness of our findings.

### 2.2. Model specification

The outcome variables of interest are the number of different types of health services delivered at each PHC. Specifically, we look at the number of deliveries (childbirths) conducted, and the average number of in-patients and out-patients treated at the PHC in the month prior to the survey. The intermediate variables of interest are related to the equipment and staff at the PHCs. The primary focal variable is the type of electricity access in the PHC. The survey contained five responses to the type of electricity access available at the PHC–regular electricity, occasional power cut, power cut in the summer, regular power cut, and no electricity. We merged the middle three responses into a single category that we labeled irregular electricity. This resulted in three categories for the electricity access variable used in our models: regular electricity, irregular electricity, and no electricity. These categories are consistent with previous research [12] and models using the original five categories do not change the substantive results. These results are shown in Table A4 in S1 Appendix. Furthermore, the availability of a functional electricity generator is also included as a secondary focal variable, since generators are most commonly used as back-up sources of electricity.

All three outcome variables considered in our analysis are count variables and exhibit over-dispersion, wherein the conditional variances ($\sigma^2$) are much greater than the conditional means ($\mu$) (Table 1). Using over-dispersed count data as dependent variables in ordinary least squares (OLS) models is known to produce biased and inefficient estimates, and instead the use of negative binomial (NB) model is recommended [16–18]. Furthermore, the number of deliveries and in-patients have a high number of zero values (Table 1), since PHCs without labor rooms would typically not be conducting deliveries and they would not keep in-patients if there are no beds in the facility. Therefore, for these two outcomes a zero-inflated negative binomial (ZINB) model was used.

The model controls for other variables that are known to influence the volume of health services delivered [16, 19, 20] analyzed here (Table 2). These covariates include the population

**Table 1. Conditional means and variances for outcome variables, and model selection.**

|  |  |  | Regular Electricity | | Irregular Electricity | | No Electricity | |  |
|---|---|---|---|---|---|---|---|---|---|
| **Dependent Variable** | **N** | **N Zeros** | **μ** | **σ²** | **μ** | **σ²** | **μ** | **σ²** | **Model Choice** |
| Deliveries | 7805 | 2744 | 21 | 2596 | 12 | 761 | 9 | 1499 | ZINB |
| In-Patient | 4540 | 1887 | 56 | 18045 | 24 | 4209 | 22 | 6760 | ZINB |
| Out-Patient | 4782 | 68 | 1248 | 2523637 | 794 | 751311 | 694 | 974745 | NB |

*Note*: *The sample size for In-Patient and Out-Patient outcome variables drop because of missing data for these two questions in the survey.*

**Table 2. List of variables used in the empirical models.**

| Variable Type | Variables |
|---|---|
| Outcome | Number of Deliveries, In-Patients, Out-Patients |
| Primary Focal Variable (FV$_P$) | Electricity (regular electricity / irregular electricity / no electricity) |
| Secondary Focal Variable (FV$_S$) | Generator availability |
| Interaction Variables (IV) | Staff availability (medical officer, lady medical officer, lady health visitor, nurse, auxiliary nurse midwife, pharmacist), Equipment availability (radiant warmer, autoclave, deep freezer, ice-lined refrigerator, centrifuge), Facility timings, medical officer residence status |
| Control Variables (CV) | Population served, Infrastructure (government building, water, toilet, beds) |
| State Fixed Effects (SFE) | Dummy variables for each State |

designated under the PHC's service area, facility timings (whether open 24x7), the number of beds, different types of staff and equipment available, and other infrastructure facilities (government building, building condition, labor room, water and toilet). Furthermore, since the state governments in India have a substantial say in the organizations and functioning of their public health systems, we expect there to be structural similarities among PHCs in each state that may vary from state to state. Accordingly, we include state fixed effects in our models. In total, we use 23 variables in our models. To test for potential multi-collinearity among these variables, we computed the Variable Inflation Factors (VIFs) for all of our model specifications and found that none of the variables had a high enough VIF to indicate significant multi-collinearity (analysis available in the code in S1 Replication materials).

## 2.3. Interaction effects

The WHO lists several potential causes that lead to deficiencies in a health system (Fig 1) [1]. While factors such as the availability of staff, equipment, alternate energy facilities (such as diesel generators), and facility timings impact the delivery of healthcare, adequate electricity access can be an underlying precursor that also affects these factors. The direct and indirect effects of electricity access are shown in Fig 2. For example, even if medical staff are available at the health center, the lack of electricity access can reduce their ability to deliver care. The hypothesis therefore is that poor electricity access has indirect effects on healthcare delivery,

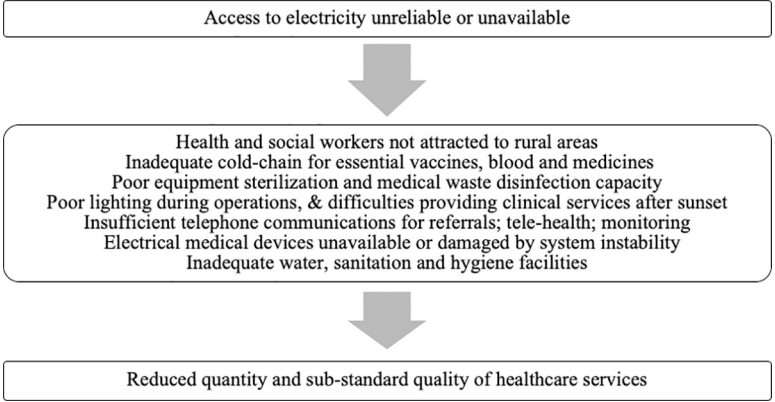

**Fig 1. Potential causes leading to deficiencies in the health system, exacerbated by poor energy access (adapted from [1]).**

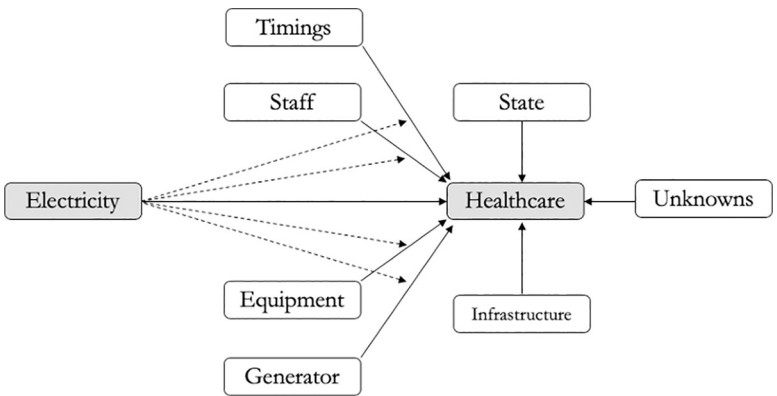

**Fig 2. Direct and indirect effects of electricity access on healthcare delivery.**

through its interaction with other variables. These interactions have therefore been included in the empirical models.

We test the robustness of our focal variable estimates in three ways. First, we check the robustness with respect to model specification by analyzing three types of models. Model 1 is a parsimonious model with only focus variables and the state fixed effects. Model 2 is an intermediate model including state fixed effects and all co-variates without interaction effects. Model 3 is a full model that includes covariates, interactions, and state fixed effects. Second, we perform sensitivity analyses to check the robustness of the estimates to any outliers in the data. Third, since our primary focal variable has three categories, we conduct three sub-sample regressions to test the robustness of the estimates by considering two categories at a time.

The final model specifications are as follows. The unit of analysis is a PHC.

*Model 1: Outcome = $\alpha_0 + \alpha_1(FV_P) + \alpha_2(FV_S) + \alpha_3(SFE) + \varepsilon$*

*Model 2: Outcome = $\beta_0 + \beta_1(FV_P) + \beta_2(FV_S) + \beta_3(IV) + \beta_4(CV) + \beta_5(SFE) + \varepsilon$*

*Model 3: Outcome = $\gamma_0 + \gamma_1(FV_P) + \gamma_2(FV_S) + \gamma_3(FV_P * FV_S) + \gamma_4(IV) + \gamma_5(FV_P * IV) + \gamma_6(CV) + \gamma_7(SFE) + \varepsilon$*

## 2.4. Limitations

Assessing the causal pathways that determine the health outcomes of a population is challenging. Many community level and individual level factors influence how people access healthcare care [21, 22]. The cross-sectional data from DLHS does not allow for including such community level fixed effects in the model. Some influential variables that impact healthcare delivery, such as the funding available to the PHC, are not available in the survey and therefore cannot be included in the model, raising the concern of potential omitted variables in our models. These data limitations may induce some bias in our estimates and preclude us from establishing a direct causal link between electricity availability and healthcare delivery. However, we are less concerned about endogeneity due to reverse causality in our focus variable. The electricity variable in DLHS measures the availability of grid electricity at the PHCs. Historically, rural electrification policies have prioritized villages with higher populations or villages with higher agricultural activity (e.g., for energizing pump sets). More recent electrification efforts have aimed for 100 percent village electrification, perhaps prioritizing villages closer to existing grid lines and then extending to villages farther away. PHCs have usually gotten electrified as part of this village electrification process. To our knowledge, though, there haven't been any

targeted policy efforts that have systematically prioritized electrification of PHCs based on healthcare demands. Thus, it is unlikely that a trend in increased healthcare demand influenced an improvement in electricity availability at PHCs. On the other hand, it is much more plausible that reliable electricity access (together with other factors) can trigger an improvement in healthcare services. That said, it may be true that villages with higher populations (which could relate to a greater number of people treated at the PHC) could have influenced better electricity availability at the respective PHCs. Accordingly, we control for population in all the models.

A second limitation of this study is the fact that the survey may not fully reflect the current scenario in India. In the seven years that have elapsed since the last round of the DLHS survey (2012–13), significant upgrades to the rural electrification infrastructure have been reported. This has decreased the number of PHCs without electricity by 41 percent, from 1919 in 2012 [23] to 795 in 2019 [13]. However, the clear variation of electricity access observed in the DLHS-4 dataset is actually an advantage for this particular analysis, since it allows us to test how that variation is associated with the level of health services delivered. Thus, our analysis offers insights regarding the fundamental connection between access to electricity and level of health services. Given that globally the delivery of healthcare services to hundreds of millions of households is still dependent on poor or no access to electricity, understanding this connection is important not just at the level of primary care in India, but also at more decentralized levels of healthcare delivery in India as well as other parts of the developing world that continue to face similar challenges. Our findings assume particular significance in light of the recent policy focus in India, as highlighted later in section 4.1.

## 3. Results

At the time of the survey, less than half of the PHCs had access to regular electricity. About 9.5 percent PHCs did not have access to any electricity at all, mostly concentrated in northern part of the country (Fig 3). About 44 percent of the PHCs reported having irregular electricity, and these were distributed across the country. The availability of essential medical equipment, staff, and the average number of basic health services delivered across PHCs with different levels of electricity access is presented in the following sections.

### 3.1. Relationship between electricity access and medical equipment

The Indian Public Health Standard [24] lays out the essential medical equipment that is necessary to deliver quality healthcare. Increasingly, much of the essential medical equipment is electricity dependent. We observe that more than 80 percent of the PHCs without electricity also did not have access to basic functional medical and diagnostic equipment such as vaccine refrigerators, deep freezers, radiant infant warmers, light microscopes, and centrifuges (Fig 4). In comparison, amongst PHCs that did have regular electricity, at least half of them had access to all of this equipment. The unavailability or dysfunctionality of essential medical equipment subsequently hampers the quality and quantity of health services that can be delivered, especially in already resource constrained contexts. Furthermore, lack of basic amenities and functional medical equipment are critical contributing factors to the rural healthcare workforce crisis in India and other countries, where doctors trained in cities are often very reluctant to serve rural postings [25].

### 3.2. Relationship between electricity access and medical staff

Ideally, a PHC is supposed to be staffed with a medical doctor, three nurses, a pharmacist, a male health worker, a female health worker, auxiliary nurse-midwives (ANMs), and other

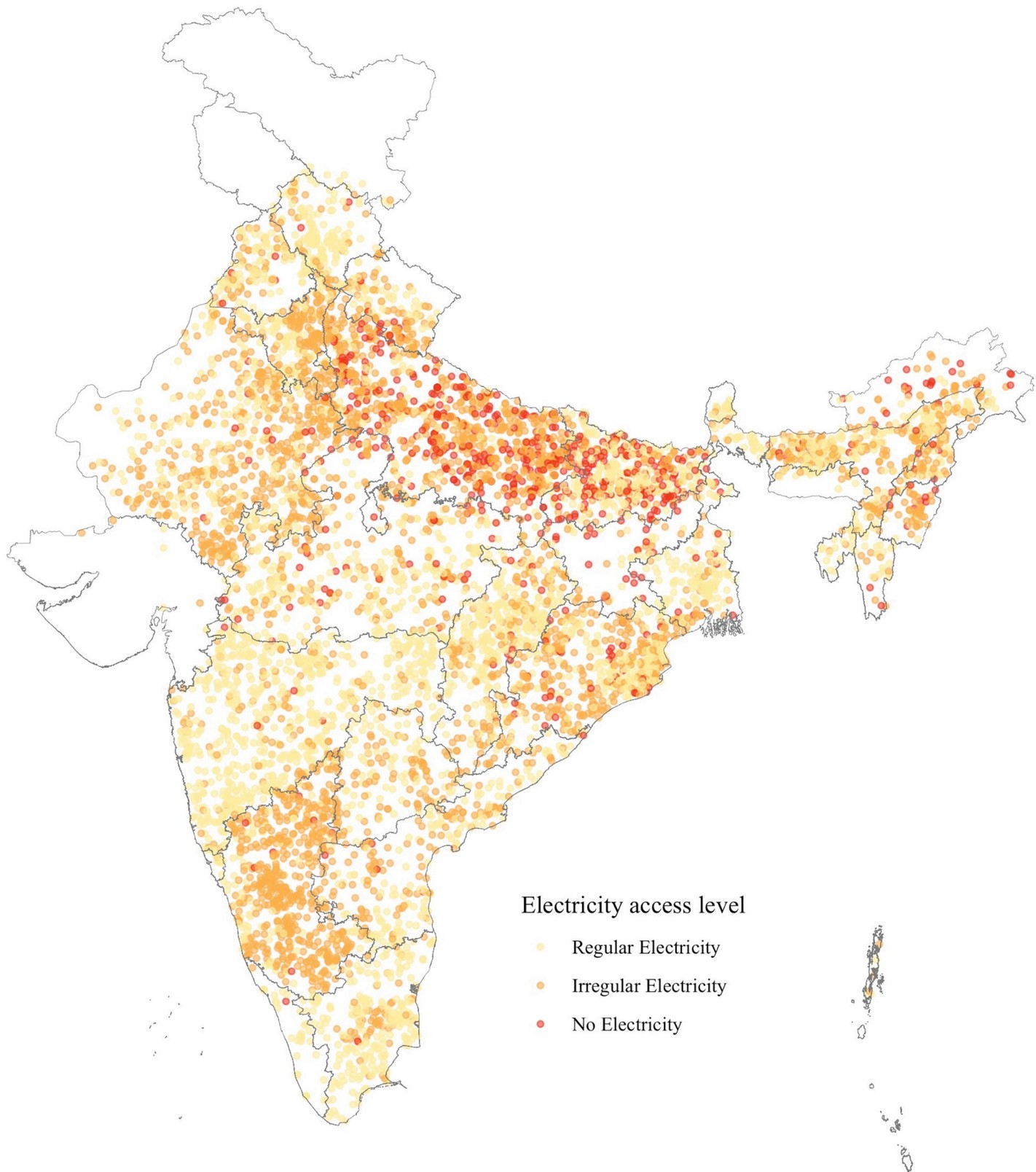

**Fig 3. Distribution of PHCs in India with different levels of electricity access at the time of survey.** Map created by authors. *Data source*: *DLHS-4. India state boundaries are reprinted from https://github.com/datameet/maps/tree/master/States under a CC BY license, with permission from DataMeet India community, original copyright 2021.*

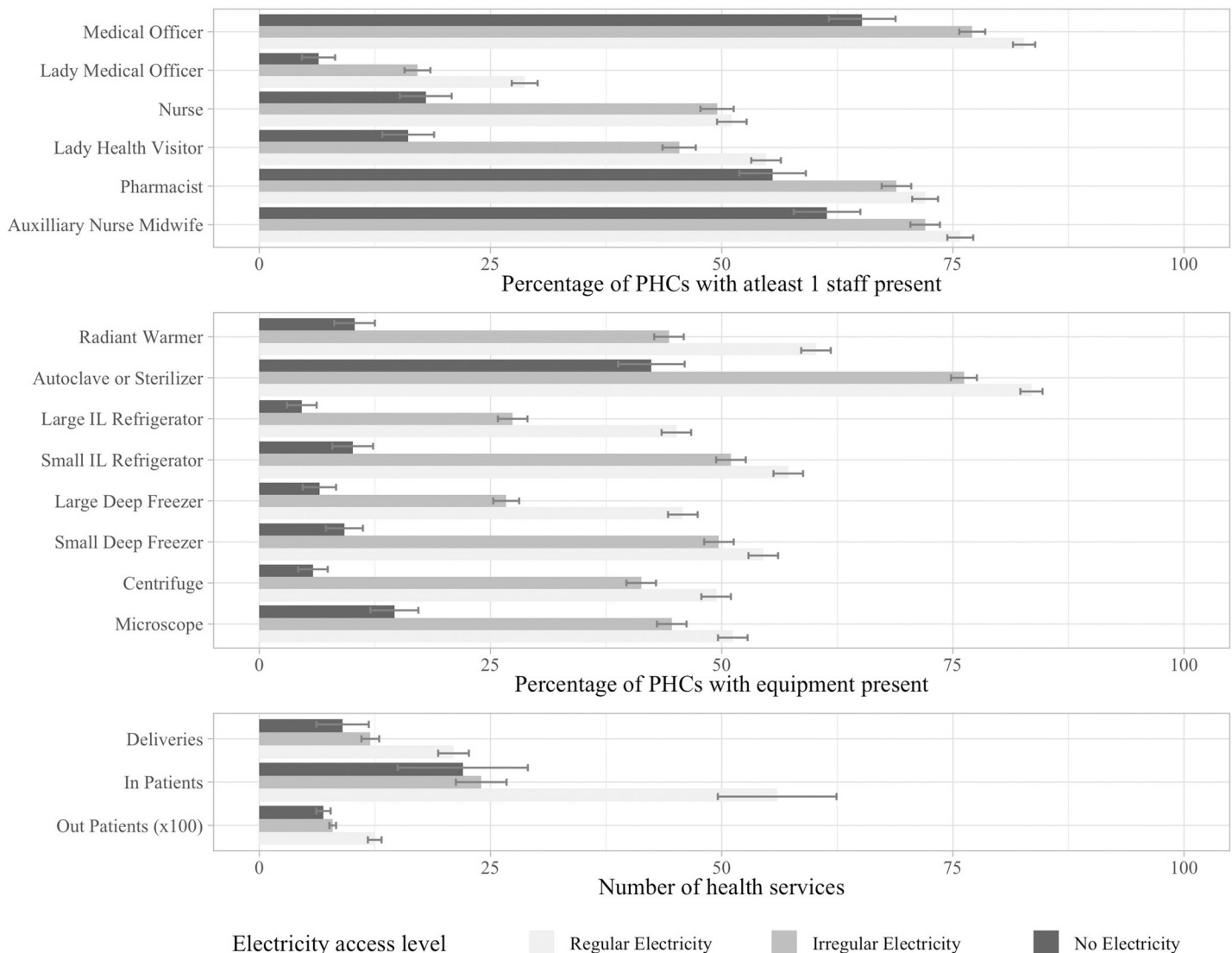

**Fig 4. (Top panel) percentage of primary health centers in which medical staff are available, (middle panel) percentage of primary health centers in which functional medical equipment is available, and (bottom panel) average number of patients per month that were treated at the primary health centers, categorized by level of electricity access.** Error bars show the 95% confidence interval for each estimate.

supporting staff [24]. Overall, the availability of medical officers, pharmacists, and ANMs was relatively high for all PHCs, hovering around 65–75 percent (Fig 4). However, across all staff categories, we find that the percentage of PHCs with at least one staff member available was lower among PHCs with lower level of electricity access. We also find a stark difference in the availability of other female staff at PHCs with electricity as compared to those without (20–30% points lower). For example, we find that while 50 percent of the PHCs with regular (and even irregular) electricity had at least one nurse available at the time of the survey, only 20 percent of the PHCs without electricity had a nurse available. This shows evidence of a gendered impact, whereby female medical staff availability is strongly associated with electricity access. This would have direct implications for expecting mothers, as some estimates have found that over 80 percent of the deliveries in the PHCs were in fact conducted by nurses [26].

## 3.3. Relationship between electricity access and healthcare delivery

Given lower equipment and staff availability at PHCs without electricity access, one would expect to see lower levels of health services. The survey data reveal that the monthly number of deliveries (childbirths) conducted, the number of inpatients, and the number of outpatients were consistently lower in PHCs with lower levels of electricity access (Fig 4). For instance, while 20 deliveries were conducted on an average in the month prior to the survey at PHCs with regular electricity, those with no electricity and irregular electricity recorded 10–12 deliveries per month on the average. Similarly, while PHCs with regular electricity treated 1250 out-patients per month on average, other PHCs treated an average of 700–800 patients per month.

For reference, IPHS guidelines classify PHCs with a load of less than 20 deliveries per month to be Type-A PHCs and those with more than 20 deliveries per month to be Type-B PHCs that would be eligible for more resources [24]. The IPHS also mentions that a PHC would at a minimum treat 960 out-patients per month. This is calculated as 40 patients per medical doctor working 6 days in a month (4 weeks). This equates to $40*6*4 = 960$ in-patients per month for each medical doctor present at the PHC. We can therefore infer than while an average PHC with regular electricity supply is able to meet the IPHS guidelines for a Type-B PHC, an average PHC with lower level of electricity access falls short of the service delivery standards. This trend of reduced service delivery and PHCs with poor electricity access is expected for at least two reasons. First, lower staff and equipment availability would reduce the health centers' capacity to treat patients in a timely manner. Second, if there is a perception among the village residents that the PHC has no electricity or very poor access, or that staff absenteeism is high, they might choose other private facilities that might be more reliable, thereby reducing the patient load at the PHC.

## 3.4. Statistical significance of observed impacts

The statistical significance of the associations reported in Sections 3.1–3.3 were tested using zero-inflated negative binomial regression models for deliveries and in-patient outcomes and negative binomial regression model for the out-patient outcome. The coefficients of the variables are reported as Incidence Rate Ratios (IRR). An IRR greater than 1 implies that an increase in the independent variable is associated with an increase in the outcome variable and vice versa.

We find that PHCs without electricity access are associated with 64 percent lower number of deliveries and 38 percent lower number of out-patients treated as compared to PHCs with regular electricity access (Table 3). The importance of having alternate electricity sources is also evident in these models, which show that *ceteris paribus* the availability of (diesel) generators is associated with 22 and 25 percent higher number of in-patients and out-patients respectively. These estimates are robust to model specification, as their direction and significance remain the same in all three specifications.

Coefficients for all dependent variables for the full model specification (model 3) are shown in Table A1 in S1 Appendix. Many of the interaction variables included in the specification in model 3 are statistically significant, indicating that the focal variable coefficients in the parsimonious model 1 and intermediate model 2 specifications that do not account for the interactions would be biased. Therefore, even though the estimates from all three model specifications are broadly consistent, in the rest of the paper we focus our interpretation on the model 3 specification, which accounts for interactions.

**3.4.1. Outlier sensitivity analysis.** With regards to the in-patient outcome variable, contrary to our expectation, in the full model specification we find that PHCs without electricity

**Table 3. Incidence Rate Ratios (IRR) (exponentiated log-odds) for the focal variables in the regression models for three outcome variables of interest.**

| Dependent Variable | Deliveries | | | In-Patient | | | Out-Patient | | |
|---|---|---|---|---|---|---|---|---|---|
| Model | Zero-Inflated Negative Binomial | | | Zero-Inflated Negative Binomial | | | Negative Binomial | | |
| Model Type | 1 | 2 | 3 | 1 | 2 | 3 | 1 | 2 | 3 |
| **Focal Variables** | | | | | | | | | |
| Irregular Electricity | 1.08* | 1.14*** | 0.97 | 0.99 | 1.05 | 1.05 | 0.87*** | 0.93*** | 0.94 |
| No Electricity | 0.52*** | 0.71*** | 0.36*** | 0.62**** | 0.90 | 1.52* | 0.61*** | 0.80*** | 0.62*** |
| Generator | 1.25*** | 1.01 | 1.03 | 1.50*** | 1.23*** | 1.22** | 1.42*** | 1.18*** | 1.25*** |
| Irregular Electricity: Generator | | | 0.92 | | | 1.02 | | | 0.89** |
| No Electricity: Generator | | | 2.10*** | | | 1.78* | | | 0.89 |
| **Control for Interactions** | No | No | Yes | No | No | Yes | No | No | Yes |
| **Control for Covariates** | No | Yes | Yes | No | Yes | Yes | No | Yes | Yes |
| **State Fixed Effects** | Yes | Yes | Yes | Yes | Yes | Yes | Yes | Yes | Yes |
| Observations | 7,805 | 7,805 | 7,805 | 4,540 | 4,540 | 4,540 | 4,782 | 4,782 | 4,782 |
| Log Likelihood | -22,799 | -22,460 | -22,416 | -14,594 | -14,419 | -14,397 | -36,114 | -35,854 | -35,832 |

Note:

*p < 0.1

**p < 0.05

***p<0.01

access are associated with 52 percent *higher* number of in-patients compared to PHCs with regular electricity access. Furthermore, this estimate is not robust to model specification, as the direction of significance reverses in the full model (model 3) compared to the parsimonious model (model 1). In order to further investigate the stability of this estimate, we conducted an outlier sensitivity analysis by sequentially excluding PHCs above certain thresholds of in-patient admission numbers (Table 4).

We find that the coefficients associated with electricity access for in-patient outcome variable are quite sensitive to a small number of PHCs that reported relatively very high number of in-patients seen at the facility. This is not surprising. Since zero inflated negative binomial data

**Table 4. Sensitivity analysis—Incidence Rate Ratios (IRR) (exponentiated log-odds) for the focal variables in the regression models for In-Patient outcome variable, under different outlier threshold conditions.**

| Dependent Variable | In-Patient (IP) | | | | | | | | |
|---|---|---|---|---|---|---|---|---|---|
| Condition | No Limit | IP<750 | IP<500 | IP<300 | IP<250 | IP<200 | IP<150 | IP<100 | IP<50 |
| **Primary Focal Variables** | | | | | | | | | |
| Irregular Electricity | 1.05 | 1.02 | 1.06 | 1.22 | 1.30** | 1.27* | **1.21** | 1.09 | 1.21 |
| No Electricity | 1.52* | 1.49* | 1.48* | 1.47* | 0.56** | 0.56** | **0.61**** | 0.79 | 0.86 |
| **Control for Interactions** | Yes | Yes | Yes | Yes | Yes | Yes | **Yes** | Yes | Yes |
| **Control for Covariates** | Yes | Yes | Yes | Yes | Yes | Yes | **Yes** | Yes | Yes |
| **State Fixed Effects** | Yes | Yes | Yes | Yes | Yes | Yes | **Yes** | Yes | Yes |
| Percent of PHCs excluded | 0 | 0.4 | 1.3 | 3.1 | 3.8 | 4.5 | **5.5** | 7.3 | 14.8 |
| Observations | 4,540 | 4,520 | 4,481 | 4,400 | 4,369 | 4,336 | **4,291** | 4,207 | 3,869 |
| Log Likelihood | -14,397 | -14,205 | -13,856 | -13,219 | -12,956 | -12,703 | **-12,346** | -11,655 | -9,461 |

Note:

*p < 0.1

**p < 0.05

***p<0.01

contain large number of zeros and the data are highly skewed towards zero, these models can be highly sensitive to the presence of extreme outliers in the data. Indeed, observations higher than the high quartile Q3 by more than three times the interquartile range (IQR) are defined as extreme outliers [27]. Based on this definition, in our sample PHCs with greater than 150 in-patient admissions can be considered as extreme outliers and thus may be justifiably excluded from the model to avoid them from severely biasing the estimates. By the same logic, any PHCs below this threshold (i.e., 150 in-patient admissions) would not be considered as extreme outliers and therefore their removal from the sample may not be justified.

Interestingly, upon further inquiry we find an external reason that supports the exclusion of part of these extreme outliers from our model. Specifically, when limiting the data to PHCs below 150 in-patient admissions per month, we notice that compared to PHCs with regular electricity access, PHCs without electricity access are associated with 39 percent lower number of in-patients (highlighted in Table 4). This results in exclusion of 249 PHCs, which equates to 5.5 percent of the PHCs for which in-patient admission data are available. Further, 172 of these 249 PHCs that reported greater than 150 inpatients were in the state of Bihar, and 16 more in the adjacent state of Uttar Pradesh. At the time of the survey, Bihar and Uttar Pradesh were at the center of an acute encephalitis outbreak, resulting in 178 deaths in 2012 alone [28]. This might explain the unusually high number of in-patients admitted at several PHCs across Bihar and the surrounding regions. Since these outbreaks were localized and did not systematically affect the entire state, these events may not be adequately captured by the state fixed effects. We further notice in Table 4 that this coefficient is stable until the exclusion of PHCs with 150 in-patient admissions, which corresponds to the extreme outlier definition.

To avoid biasing our estimates for the purpose of this analysis, we believe there is reasonable justification for excluding the 249 PHCs with greater than 150 monthly in-patient admissions from the data used in modeling the in-patient outcome variable. Corresponding to this specification, PHCs without electricity access are associated with 39 percent lower number of in-patients treated as compared to PHCs with regular electricity access. We conducted similar analyses for the other two outcome variables (deliveries and out-patients) and found that the estimates for PHCs without electricity are robust and not sensitive to outliers. Excluding the outliers for these models did not substantially change the coefficient estimates and the findings. These results are shown in Tables A2 and A3 in S1 Appendix.

**3.4.2. Effect of interactions.** The moderating effect of electricity access on the other variables can also be observed from the statistically significant interactions in the full model results shown in Table A1 in S1 Appendix. Consider two specific examples. First—while the availability of each additional lady health visitor was associated with 8 percent lower number of deliveries, when there is no electricity availability of each additional lady health visitor was associated with 13.5 percent higher number of deliveries. Interaction takes a multiplicative effect and the calculation is as follows: coefficient of *Lady Health Workers* (0.08) x coefficient of *No Electricity*: *Lady Health Workers* (1.69) = 0.135 = 13.5%. Refer to Appendix for coefficients. This suggests that staffing adequate lady health workers especially in health centers without electricity could enhance rural women's access to delivery services. Second–while having a generator does not seem to have a significant impact on the number of deliveries conducted overall, when there is no electricity having a generator was associated with doubling in the number of deliveries. This highlights the criticality of non-grid power sources in rural health facilities with deficient access to grid power.

**3.4.3. Sub-sample regressions.** One surprising finding in the fully specified models for all three outcome variables is the absence of any statistically significant difference in the outcomes for PHCs with irregular electricity compared to those with regular electricity (Table 3). Having unreliable ("irregular") access to electricity is also expected, *ex-ante*, to affect the functioning

**Table 5. Sub-sample regressions: Incidence Rate Ratios (IRR) (exponentiated log-odds) for the focal variables in the regression models for three outcome variables of interest, tested with two categories at a time.**

| Dependent Variable | Deliveries | In-Patient | Out-Patient |
|---|---|---|---|
| Model | Zero-Inflated | Zero-Inflated | Negative Binomial |
| | Negative Binomial | Negative Binomial | |
| **Reference: Regular Electricity** | | | |
| No Electricity | 0.31*** | 1.69** | 0.64*** |
| **Reference: Irregular Electricity** | | | |
| No Electricity | 0.45*** | 1.10 | 0.70*** |
| **Reference: Regular Electricity** | | | |
| Irregular Electricity | 0.93 | 1.07 | 0.93 |
| **Control for Interactions** | Yes | Yes | Yes |
| **Control for Covariates** | Yes | Yes | Yes |
| **State Fixed Effects** | Yes | Yes | Yes |

Note:

*p < 0.1

**p < 0.05

***p<0.01

of equipment and subsequently could lower the health services delivered at these PHCs. We expected the coefficient on irregular electricity to be statistically significant even if more muted compared to PHCs without any electricity, but we did not find this relationship.

Thus, in order to test the robustness of our estimates further, we conducted 3 sub-sample regressions each with two categories of electricity access at a time (Table 5). For example, in the first sub-sample regression, we exclude all PHCs which have "Irregular Electricity" and estimate the coefficient for "No Electricity" with reference to PHCs with "Regular Electricity" under the full model specification. In the other two sub-sample regressions, we similarly exclude PHCs with "Regular Electricity" and "No Electricity" respectively and compare the remaining two categories.

We found the results from the sub-sample regressions to be substantively consistent with the overall model. PHCs without electricity are associated with substantially lower number of deliveries and out-patients compared to PHCs with regular as well as irregular electricity. However, we observe no statistically significant difference between PHCs with regular and irregular electricity access. Even when we expand the analysis using the original five categories for the level of electricity access, the only additional significant result vis-à-vis regular electricity access is with regard to the number of out-patients at PHCs with regular power cuts (see Table A4 in S1 Appendix). Overall, we do not find significant relationships with regard to the PHCs with irregular electricity supply compared to those with a regular supply. This may be related to the reliability of the irregular electricity categories in the electricity variable. For example, in two PHCs experiencing the same quality of electricity supply, one respondent may answer "occasional power cut" whereas the other may answer "regular power cut", depending on their relative contexts. This may introduce subjectivity in how the irregular electricity variable is coded, potentially masking the significance of certain types of irregularity. Future studies can gather more accurate electricity reliability data complemented with qualitative studies to offer a more nuanced understanding of the relationship between unreliable electricity access and health service delivery.

**3.4.4. District fixed effects.** Our use of state fixed effects is motivated by a constitutional provision of the Government of India, which grants states the legislative authority over matters

of public health. However, state fixed effects may not sufficiently account for the unobserved spatial heterogeneity within the states, for example, as observed in the outlier sensitivity analysis in section 3.4.1. We therefore compared the state fixed effect model to a district fixed effect model. There are 536 districts in our data and the zero-inflated negative binomial models did not converge because of such high-dimensional fixed effect. Instead, for deliveries and in-patient outcomes we use OLS regressions to compare the two fixed effect models. We observe that the district fixed effect models have a slightly higher explanatory power compared to the state fixed effect models, as evident from the slight increase in the adjusted $R^2$ values (Table 6). However, more importantly, the focal coefficients for models of all three outcome variables are robust to the type of fixed effect (state or district) specified. Specifically, the type of fixed effect specified does not substantially alter the magnitude, direction, or significance of the focal coefficients (Table 6).

**3.4.5. Pooled data from DLHS-3 survey.** The analysis presented in this paper is based on the facility-level data from the fourth round of the District Level Household and Facility Survey (DLHS-4) conducted in 2012–13. While this survey has since been discontinued, facility-level data was collected in only one previous round of the survey–DLHS-3 –in 2007–08. Had the two surveys covered the same PHCs, the resulting panel data would have allowed for the inclusion of PHC-level fixed effects, potentially enabling a stronger control of time-invariant facility-level characteristics. However, DLHS-3 and DLHS-4 were based on different sampling units and therefore survey different PHCs in each round. Still, pooling the data from both the surveys to the extent possible can help further test the robustness of the estimates, while introducing a time fixed effect.

We therefore created a pooled dataset and analyzed the full model specification with two-way fixed effects (Table 7). A stand-alone analysis for DLHS-3 data is also included. Consistent with all the previous results, we observe that the coefficients for PHCs with no electricity are robust to the dataset specification for the delivery and out-patient outcome variables. Further, the analysis of DLHS-3 data shows that compared to PHCs with regular electricity, those without electricity access were associated with 51 percent lower number of in-patient admissions,

**Table 6. OLS estimates and Incidence Rate Ratio (IRR) (exponentiated log-odds) for the focal variables in the regression models with state and district fixed effects.**

| Dependent Variable | Deliveries | | In-Patient | | Out-Patient | |
|---|---|---|---|---|---|---|
| Model | OLS | | OLS | | Negative Binomial | |
| Fixed Effect Type | State FE | District FE | State FE | District FE | State FE | District FE |
| **Focal Variables** | | | | | | |
| Irregular Electricity | 2.68 | 2.14 | 20.49*** # | 10.47* # | 0.94 | 0.97 |
| No Electricity | -15.51*** | -17.46*** | 16.75** # | 13.98* # | 0.62*** | 0.68*** |
| Generator | 1.84 | 1.86 | 7.71** | 5.89 | 1.25*** | 1.23*** |
| **Control for Interactions** | Yes | Yes | Yes | Yes | Yes | Yes |
| **Control for Covariates** | Yes | Yes | Yes | Yes | Yes | Yes |
| **State Fixed Effects** | Yes | No | Yes | No | Yes | No |
| **District Fixed Effects** | No | Yes | No | Yes | No | Yes |
| Adjusted $R^2$ | 0.49 | 0.53 | 0.57 | 0.66 | - | - |
| Log Likelihood | | | | | -35,832 | -36,421 |

Notes:

*p < 0.1

**p < 0.05

***p<0.01

# The models for in-patient outcome in this table do not exclude the outliers (see section 3.4.1)

**Table 7. Incidence Rate Ratios (IRR) (exponentiated log-odds) for the focal variables in the regression models with different dataset specifications.**

| Dependent Variable | Deliveries | | | In-Patient | | | Out-Patient | | |
|---|---|---|---|---|---|---|---|---|---|
| Model | Zero-Inflated | | | Zero-Inflated | | | Negative Binomial | | |
| | Negative Binomial | | | Negative Binomial | | | | | |
| Dataset Type | DLHS 3 only | DLHS 4 only | DLHS 3 & 4 | DLHS 3 only | DLHS 4 only | DLHS 3 & 4 | DLHS 3 only | DLHS 4 only | DLHS 3 & 4 |
| **Focal Variables** | | | | | | | | | |
| Irregular Electricity | 0.76 | 0.97 | 1.00 | 0.47*** | 1.05 | 0.92 | 0.88 | 0.94 | 0.95 |
| No Electricity | 0.33*** | 0.36*** | 0.58*** | 0.49*** # | 1.52*# | 1.23 # | 0.71*** | 0.62*** | 0.69*** |
| Generator | 0.94 | 1.03 | 1.00 | 1.13 | 1.22** | 1.18*** | 0.99 | 1.25*** | 1.14*** |
| **Control for Interactions** | Yes | Yes | Yes | Yes | Yes | Yes | Yes | Yes | Yes |
| **Control for Covariates** | Yes | Yes | Yes | Yes | Yes | Yes | Yes | Yes | Yes |
| **State Fixed Effects** | Yes | Yes | Yes | Yes | Yes | Yes | Yes | Yes | Yes |
| **Time Fixed Effect** | No | No | Yes | No | No | Yes | No | No | Yes |
| Observations | 7,096 | 7,805 | 14,898 | 7,023 | 4,540 | 11,563 | 7,214 | 4,782 | 11,996 |
| Log Likelihood | -23,675 | -22,416 | -46,967 | -22,090 | -14,397 | -36,763 | -55,589 | -35,832 | -91,789 |

Notes:

*p < 0.1

**p < 0.05

***p<0.01

# The models for in-patient outcome in this table do not exclude the outliers (see section 3.4.1)

which is in line with our expectation. This finding lends further credence to the outlier sensitivity analysis presented in section 3.4.1, and our hypothesis that localized outbreaks of encephalitis in 2012 may have resulted in disproportionately higher in-patient admissions in some of the PHCs as reported in DLHS-4. We also observe from the DLHS-3 model that compared to PHCs with regular electricity, even PHCs with irregular electricity were associated with 53 percent lower number of in-patient admissions, which is something we don't observe in the DLHS-4 model. As discussed in section 3.4.3, a more nuanced definition of irregular electricity is needed to understand its association with healthcare delivery.

In summary, after controlling for a number of known factors including staff, equipment, and other infrastructure, lack of electricity access is statistically associated with reduced levels of healthcare services at primary health centers in India. The findings, particularly for deliveries and out-patient outcome variables, are consistent and robust to model and data specifications. Furthermore, based on a series of robustness checks and sensitivity analyses we have presented, we find that this main result also holds for in-patient outcomes.

## 4. Discussion

### 4.1. Energy and equitable access to healthcare

Equitable access to healthcare facilities depends on a number of supply and demand factors. There are four key supply-side determinants–resource allocation, physical access, human resources, and technology [29]. In addition to these factors, the quality of care delivered at public health facilities also determines whether people will choose to access these facilities or choose more expensive private options. In studies on supply-side determinants and the quality of healthcare services, access to electricity is rarely discussed, or is mentioned in a limited context of ensuring infrastructure availability. In reality, reliable electricity directly or indirectly affects much of the supply-side and quality factors of healthcare delivery.

The analysis presented in this paper provides new empirical evidence to the previously supposed notions that lower levels of energy access is associated with lower provision of health services and with lower availability of medical staff and functional equipment at rural primary health centers. By clearly establishing that fundamental linkage empirically, this paper makes a strong case for rural health system planners and government health departments to pay much more attention to understanding and integrating reliable energy access as an enabler of more equitable access to primary healthcare. For example, one of the key pillars of *Ayushmaan Bharat*–the flagship health policy of the Government of India–is the development of Health Sub-Centers as Health and Wellness Centers (HWCs). HWCs are envisioned to deliver more comprehensive care, shifting some of the functions downward from the PHCs [30]. As of 2019, 26.3 percent of the 149,590 rural Sub-Centers in India were without access to electricity [13]. These Sub-Centers cumulatively serve more than 200 million rural individuals, assuming each sub-center serves 5,000 individuals. Given that the HWCs will be more decentralized and spread out than PHCs, in view of the evidence provided in this paper it is imperative that reliable electricity access be considered an integral part of the strategy for developing HWCs. These findings are also relevant to other parts of the developing world where unreliable electricity at public health facilities still affects hundreds of millions of poor and underserved communities.

## 4.2. Gendered impacts

Studies on gender-based inequalities in healthcare have documented how women, especially those associated with lower socio-economic status have experienced higher prevalence of morbidity and have generally had lower levels of utilization of healthcare as compared to men [31]. Recent studies have also highlighted gender disparities in household energy use [32]. Findings in this paper show how the lack of energy access at rural health centers further aggravates these inequalities. Poor energy access significantly affects a mother's ability and experience of safe childbirth. In the empirical context of this paper, most of the deliveries happen in the evenings or at night [33], and the lack of electricity access means providing even basic lighting for the delivery becomes a challenge, let alone the ability to handle complications.

The substantially strong negative association of electricity deficit with the number of deliveries indicates that poor electricity access is disproportionately associated with adverse health services for women and newborns. In the absence of reliable public health facilities many women resort to delivering at private facilities that are often farther away and more expensive, further increasing their burden. In the absence of functional emergency care services at public health facilities, expectant mothers with potential complications are driven to private health facilities, where the chance of undergoing a caesarian section is three times more likely as compared to public health facilities [34]. Furthermore, by directly or indirectly leading to poor living and working conditions and poor safety, lower energy access acts as a barrier for attracting and retaining female medical staff at the health centers, which in turn makes it more difficult for women to access safe and timely care. Therefore, poor energy access disproportionately affects women's access to safe and quality healthcare. This provides yet another compelling reason why bridging the energy gap is critical.

## 4.3. Decentralized renewable energy solutions

In addition to the mere availability of energy, the reliability of energy supply is also important for providing timely and quality care. In most health centers, the reliability gap is bridged using alternate power sources such as diesel generators and inverters. We have noted in our analysis that electricity deficit PHCs with generators conducted twice as many deliveries

compared to those without a generator. The importance of these alternate power sources in improving the health services is therefore reinforced from our findings. However, the use of diesel generators (DGs) at these rural health facilities presents its own set of challenges [11]. DGs pollute the local environment, produce a lot of noise which affects the care environment, they break down regularly, repairs take time, and in remote centers, the very procurement and storage of adequate quantities of fuel on a regular basis is a challenging task [35]. With a number of competing needs that demand the limited financial resources available to these health centers, PHC managers (typically doctors) end up rationing and saving fuel for very critical and emergency cases.

Alternatively, there are a number of innovative examples that leverage decentralized renewable energy technologies to improve service delivery at rural health centers. The use of decentralized solar technologies to improve healthcare delivery has gained particular prominence among practitioners globally. In India alone, over one thousand PHCs in states like Maharashtra, Chhattisgarh, Tripura and Karnataka have been solar powered through collaborations between state and local governments and external agencies [36–38]. Preliminary evaluation studies of decentralized renewable energy based solutions have shown that these interventions do catalyze positive outcomes with regard to service availability (especially at night), day-to-day operations, staff retention, and community satisfaction [5, 37, 39]. These studies further show that other health system and implementation factors need to be aligned in order for the energy interventions to sustain the positive health outcomes [39].

In light of the findings from this paper and given the urgency of SDG 3 and SDG 7, there is a need to not only scale up these demonstrated models, but also improve the research and development focus on developing new innovative models that leverage point-of-care technologies enabled by decentralized renewable energy to ensure more people in disadvantaged regions have access to quality healthcare.

## Supporting information

**S1 Appendix.**
(DOCX)

**S1 Replication materials.**
(ZIP)

## Acknowledgments

The authors would like to thank Cale Reeves, Ariane Beck, Mark Hand, John Cornwell, Chandler Stolp and two anonymous reviewers for helpful discussions and suggestions. We are also thankful for the feedback from participants at the 2$^{nd}$ International Conference on Energy Research and Social Science. All remaining errors are ours alone.

## Author Contributions

**Conceptualization:** Vivek Shastry, Varun Rai.

**Data curation:** Vivek Shastry.

**Formal analysis:** Varun Rai.

**Methodology:** Vivek Shastry, Varun Rai.

**Software:** Vivek Shastry.

**Supervision:** Varun Rai.

**Writing – original draft:** Vivek Shastry.

**Writing – review & editing:** Varun Rai.

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
