## [Decision Letter · Decision Letter 0]

8 Dec 2020

PONE-D-20-28579

How Energy Access Impacts Primary Healthcare

PLOS ONE

Dear Dr. Shastry,

Thank you for submitting your manuscript to PLOS ONE. After careful consideration, we feel that it has merit but does not fully meet PLOS ONE’s publication criteria as it currently stands. Therefore, we invite you to submit a revised version of the manuscript that addresses the points raised during the review process.

Your revisions should address the specific critiques related to the regression modelling, including the endogeneity and model specification issues raised by Reviewer 1 and the possibility of multi-collinearity and the potential need for interaction terms and sub-sample regression raised by Reviewer 2.

We look forward to receiving your revised manuscript.

Kind regards,

Hisham Zerriffi

Academic Editor

PLOS ONE

Journal Requirements:

2. Please consider modifying your title to ensure that it is specific, descriptive, concise, and comprehensible to readers outside the field (for example by including the name of the country where the study took place ).

'..VR acknowledges support from the Elspeth Rostow Memorial Fellowship...'

'The authors received no specific funding for this work.'

5. Please include captions for your Supporting Information files at the end of your manuscript, and update any in-text citations to match accordingly. Please see our Supporting Information guidelines for more information: http://journals.plos.org/plosone/s/supporting-information

6. We note that Figure 3 in your submission contains map images which may be copyrighted.

We require you to either (a) present written permission from the copyright holder to publish these figure specifically under the CC BY 4.0 license, or (b) remove the figure from your submission:

a. You may seek permission from the original copyright holder of Figure 3 to publish the content specifically under the CC BY 4.0 license. 

b. If you are unable to obtain permission from the original copyright holder to publish this figure under the CC BY 4.0 license or if the copyright holder’s requirements are incompatible with the CC BY 4.0 license, please either i) remove the figure or ii) supply a replacement figure that complies with the CC BY 4.0 license. Please check copyright information on all replacement figures and update the figure caption with source information. If applicable, please specify in the figure caption text when a figure is similar but not identical to the original image and is therefore for illustrative purposes only.

Reviewers' comments:

Reviewer's Responses to Questions

**Comments to the Author**

1. Is the manuscript technically sound, and do the data support the conclusions?

Reviewer #1: Partly

Reviewer #2: Yes

2. Has the statistical analysis been performed appropriately and rigorously? 

Reviewer #1: No

Reviewer #2: Yes

3. Have the authors made all data underlying the findings in their manuscript fully available?

Reviewer #1: Yes

Reviewer #2: Yes

4. Is the manuscript presented in an intelligible fashion and written in standard English?

Reviewer #1: No

Reviewer #2: Yes

5. Review Comments to the Author

Reviewer #1: How Energy Access Impacts Primary Healthcare

PONE-D-20-28579

Summary

-------

This paper reports results from cross-sectional analyses of data on primary health centers (PHCs) from one round of India's District Level Household and Facility Survey. Specifically, using zero-inflated negative binomial (ZINB) and negative binomial (NB) regressions, the authors explore the relationship between a set of PHC-level healthcare use and quality metrics in the month prior to the survey, and PHC-level electrification status. The authors conclude that lack of access to electricity is associated with (i) lower availability of essential medical equipment; (ii) lower availability of essential medical staff; and (iii) lower levels of healthcare service delivery (number of childbirths, in-patient visits, and outpatient visits) at the PHC level.

Main comments

-------------

1. Spatial variation in access to electricity is endogenously driven by a host of unobserved regional, institutional and socioeconomic characteristics. These unobserved characteristics can and do induce variation in the composition of households/beneficiaries, investments in complementary infrastructure (such as rural roads), and the presence or absence of other social welfare schemes. All of these can--and do--independently induce variation in the availability, quality, staffing and use of healthcare facilities. Beyond the inclusion of state fixed-effects, the authors have made no effort to account for this inherent endogeneity, which is surprising given the relative richness of their data. Extensions of the existing analyses using basic quasi-experimental approaches that take advantage of the metrics on PHS characteristics available (e.g., propensity score matching or other matching methods; two-step Heckman correction/endogenous treatment regression models) would greatly enhance the analytical rigor of the paper and allow the authors to make a more significant contribution to the literature on energy and healthcare delivery. Absent the use of these additional approaches, the simple cross-sectional regressions presented in the study (even those that control for a host of PHS-level characteristics) are insufficient to support the causal claims that the authors make throughout the paper (e.g., "how energy access impacts primary healthcare" in paper title; "effect on" medical equipment/medical staff/health service delivery in section 3.1/3.2/3.3).

2. Various aspects of the data and empirical specification are unclear or seemingly arbitrary.

(I) why does the sample of PHCs fall by nearly 50% when looking at number of in-patient and outpatient visits (Table 1)?

(II) Why would an acute encephalitis outbreak that affected only one state (Bihar, p. 13) necessitate the removal of hundreds of PHCs from the analytical sample if the empirical specification also contains state fixed-effects (p. 6), which control for all unobserved state-specific shocks in a cross-sectional analysis? It is unclear why the authors feel that this adjustment to the sample (which reverse a statistically significant result) is justified while "any exclusions beyond this results in too many unjustified exclusions" (p. 14).

(III) What additional value do the ZINB and NB add relative to a conventional linear regression model? Without benchmark OLS estimates, seeing how the ZINB/NB estimates provide more precise results is difficult.

(IV) Similarly, how do the inclusion of dozens of controls and interactions affect the coefficients on the main "focus variables" (FVs)? Without benchmark results included from a parsimonious specification (with only the FVs and the state fixed-effects), the stability (or lack thereof) of the estimates of interest is unclear.

(V) Why do the authors use state fixed-effects instead of district fixed-effects? Presumably the District Level Household and Facility Survey contains information on the district where each PHC is located, and district fixed-effects would control for unobserved spatial characteristics more rigorously.

Additional comments

-------------------

3. "did not identify a single study in which linking energy access and health outcomes was the primary objective” - This needs to be clarified to indicate that it refers to access to electricity, specifically. SDG7 defines energy access in terms of access to electricity as well as access to clean cooking, and there is a broad literature connecting access to cleaner cooking solutions with health outcomes (see Jeuland & Pattanayak, 2012).

4. "The analysis presented in this paper provides robust empirical evidence..." (p. 3) - Why is it "robust"? Robust to what? Aside from sensitivity analyses relating to only the in-patient visits indicator, the paper does not present robustness tests for the main results.

5. Why does the paper rely only on one round of the DLHS? Mani et. al. (2019), who the paper cites, appear to use both DLHS-3 as well as DLHS-4. Would the use of multiple rounds allow for a PHC- or district-level panel/repeated cross-section to be created? This would enable more sophisticated analyses that control for trends in electrification over time, which the authors also note are a concern.

6. If urban PHCs are completing the same survey instrument for the DLHS (i.e., if they are reporting their access to electricity using the same five responses used by the rural PHCs), it is not immediately clear why they need to be removed from the analytical sample. Inclusion of a rural/urban fixed-effect in the main specification should account for unobserved context-specific differences between the two types of PHCs.

7. "...models using the original five categories do not change the substantive results." (p. 5) - These results should also be presented in the paper (in an appendix).

8. "However, we are not concerned about endogeneity due to reverse causality in our focus variable, as the quantity of health service delivered in the prior month at the health center does not cause changes in the level of electricity access." (p. 8) - As noted in main comment #1, endogeneity is not limited to concerns about reverse causality. Some third, unobserved variable can independently induce variation in both the FVs as well as electricity access (omitted variable bias), and the current analysis does not capture that.

9. Why is "generator" a focus variable in Table 3 but not indicated as such in Table 2?

10. The World Health Organization uses the term "female health worker" (see https://www.who.int/news-room/commentaries/detail/female-health-workers-drive-global-health), not "lady health worker," which is used throughout the paper.

11. "The importance of these alternate power sources in improving the health services is reinforced from our findings" (p. 18) - Please clarify how the paper's current findings reinforce the need for additional solar deployment. The previous section highlights the importance of lighting at night ("most of the deliveries happen in the evenings or at night" - p. 18), and intermittent PHS-level solar panels do not address that concern.

12. Typos:

- "Medical equipment that is..." not "medical equipment are" (used twice on p. 10)

- "consistently lower" not "consistently lesser" (p. 11)

References

----------

Jeuland, M. A., & Pattanayak, S. K. (2012). Benefits and Costs of Improved Cookstoves: Assessing the Implications of Variability in Health, Forest and Climate Impacts. PLoS ONE, 7(2), e30338. https://doi.org/10.1371/journal.pone.0030338

Mani S, Patnaik S, Dholakia HH. State of Electricity Access for Primary Health Centres in India, - Insights from the District Level Household and Facility Survey (DLHS-3 and DLHS-4) [Internet]. New Delhi: CEEW; 2019 Feb [cited 2019 Oct 18]. Available from: https://www.ceew.in/sites/default/files/CEEW-The-State-of-Electricity-Access-for- Primary_0.pdf

Reviewer #2: The paper is based on the nationally representative District Level Household Survey (DLHS-4) data and talks about the importance of electricity access for healthcare services. It is generally well written and relevant to PLOSONE readership, especially in the decade of action for attainment of access to energy for community services. I recommend acceptance for publication after addressing the following points:

1. Since the DLHS data is around a decade old, authors should continuously refer to the latest data (say, Rural Health Statistics or any other such data on cold chain services or appliance inventory or electrification status), wherever possible. This will make it more relevant to the current context in India. For instance, somewhere in the Introduction section, authors should also highlight the current situation of PHC electrification through the latest rural health statistics (RHS) data. That will show how has the situation of unelectrified PHCs changed in the last 7-8 years. Currently, there are only around 5% of the PHCs that are unelectrified in the country, and since the process of household electrification is over, the government should try and electrify these facilities as soon as possible.

2. In Table 1 (page 5), in order to put these numbers into perspective, authors should also highlight Indian Public Health Standards (IPHS) numbers that a PHC is supposed to serve in Table 1. For instance, what does a number of 9.65 deliveries mean? Further, how many PHCs are able to meet the IPHS guidelines for Deliveries, Inpatient and Out-Patient?

3. There will be massive scope for multicollinearity among the independent variables, as authors themselves highlight in their paper. For example, "much of the essential medical equipment are electricity dependent" and "percentage of PHCs with at least one staff member available was lower among 11 PHCs with a lower level of electricity access." While specifying the model on page 6, the authors should also talk explore and discuss the degree of multicollinearity between the independent variables (say, through VIF). This will help in both getting rid of redundant variables and making the effect sizes seem more robust.

4. When the authors say "Out Patient services decrease noticeably as well in PHCs with irregular electricity supply" on page 11, I am assuming it comes from the results. If indeed that is the case, please refer to the Table number here. Also, authors should be careful while making such statements. It is not clear if Out-patient services decrease with poor electricity supply. You would need a time-series data to use the word 'decrease'. This is cross-sectional data. At best, it can only be said that PHCs with irregular electricity supply can serve a lesser number of OPD patients.

5. Visualization comment - In Figure 4 (page 12), authors should use a different colour for PHCs with regular electricity supply (maybe lighter grey).

6. In page 14, authors mention “We conducted similar analyses for the other two outcome variables (deliveries and out-patients) and found that these models are not as sensitive to the outliers.” Just like Table 4, even these results on sensitivity analysis for deliveries and outpatients should be presented in the appendix.

7. Authors should also include the result on the interaction of electricity supply from the grid with the availability of generator in Table 3. For example, an irregularly electrified PHC with a generator is better than another irregularly electrified PHC without a generator. In remote regions where it is challenging to extend electricity supply through the grid (or where grid fluctuations are higher), this result will be important to make a case for electricity backups.

8. In Table 3 (page 15), there is no significant difference between PHCs with irregular electricity and regular electricity supply. What policy message can we derive out of this for the current context? Are you saying that even if you provide irregular electricity supply, it is not going to significantly hamper the service delivery? Some discussion on this would be useful. I think in order to make this analysis even more policy-relevant, authors should perform following three separate regressions apart from the one presented in Table 3 (even if they choose to show these results in the Appendix). This will also help in doing the robustness checks for their results:

a. A subsample regression analysis only between irregularly electrified and regularly electrified PHCs to see if you can see the difference now. If there is a significant difference, then please try to highlight why this difference got lost in the bigger model where you had all three categories together (in Table 3).

b. Again, a sub-sample regression between irregularly and not electrified PHCs to see how service deliveries differ between these two categories.

c. Since the problem of no electricity access for PHCs is largely going to be done away with soon, club PHCs with irregular electricity supply and no electricity supply (assuming that PHCs with irregular power supply are equally worse).

These regressions might help in coming up with certain important policy recommendations.

9. In page 15, I think there is something wrong with the sentence “Absent reliable public health

facilities many women resort to delivering at private facilities that are often farther away and

more expensive, further increasing their burden.”

10. In page 16, authors write “Second – while having a generator does not seem to have a significant impact on the number of deliveries conducted overall when there is no electricity having a generator was associated with doubling in the number of deliveries. This highlights the criticality of non-grid power sources in rural health facilities with deficient access to grid power.” I believe these results on access to electricity through generators should also be shown in Table 3, as this is equally relevant.

11. About page 18, section 4.3 – Decentralized Renewable Energy (DRE) may not be a panacea and may also have its own challenges. In many places, they may even not have worked out well. If the DRE systems fail, it will have substantial cost implications. Some discussion on that aspect would also be useful to make a more balanced point around DRE. How to make DRE a more workable solution?

6. PLOS authors have the option to publish the peer review history of their article (what does this mean?). If published, this will include your full peer review and any attached files.

Reviewer #1: No

Reviewer #2: **Yes: **Sunil Mani

---

## [Author Response · Author response to Decision Letter 0]

9 Feb 2021

Dear Editors,

In addition to the changes made in the revised manuscript that have been summarized in the response to reviewers document, we have made the following changes as required by the journal and mentioned in the decision letter emailed by the academic editor. 

1. Edited the manuscript to meet PLOS ONE's style requirements.

2. Title has been modified to make it more specific and descriptive.

3. Fellowship information has been removed from the acknowledgements section.

4. Amended the abstract on the online submission to make it identical with the manuscript.

5. Supporting information captions have been included in the manuscript.

6. A different shapefile with CC BY 4.0 license has been used in the map in Figure 3 and a written permission has been uploaded.

7. All figures have been uploaded and verified with PACE. Figures have been uploaded as separate files and removed from the within the manuscript.

Regards,

Vivek

---

## [Decision Letter · Decision Letter 1]

15 Apr 2021

PONE-D-20-28579R1

Reduced health services at under-electrified primary healthcare facilities: Evidence from India

PLOS ONE

Dear Dr. Shastry,

Thank you for submitting your manuscript to PLOS ONE. My apologies for the delay in the decision on your manuscript. During this time I am finding that reviewers require some extra time to complete reviews and prefer to provide the extra time rather than seek new reviewers.

After careful consideration, we feel that it has merit but does not fully meet PLOS ONE’s publication criteria as it currently stands. Therefore, we invite you to submit a revised version of the manuscript that addresses the points raised during the review process.

While one reviewer is satisfied by the prior revision, one reviewer still views the manuscript as having methodological issues that are not well addressed by the last round of reviews. Upon consideration of the manuscript and the reviewer comments, I am in agreement and would like to see a revised manuscript that either implements at least some of the suggested changes by the reviewer and provides strong justification for any changes that are not made.

We look forward to receiving your revised manuscript.

Kind regards,

Hisham Zerriffi

Academic Editor

PLOS ONE

Reviewers' comments:

Reviewer's Responses to Questions

**Comments to the Author**

1. If the authors have adequately addressed your comments raised in a previous round of review and you feel that this manuscript is now acceptable for publication, you may indicate that here to bypass the “Comments to the Author” section, enter your conflict of interest statement in the “Confidential to Editor” section, and submit your "Accept" recommendation.

Reviewer #1: (No Response)

Reviewer #2: All comments have been addressed

2. Is the manuscript technically sound, and do the data support the conclusions?

Reviewer #1: Partly

Reviewer #2: Yes

3. Has the statistical analysis been performed appropriately and rigorously? 

Reviewer #1: No

Reviewer #2: Yes

4. Have the authors made all data underlying the findings in their manuscript fully available?

Reviewer #1: Yes

Reviewer #2: Yes

5. Is the manuscript presented in an intelligible fashion and written in standard English?

Reviewer #1: Yes

Reviewer #2: Yes

6. Review Comments to the Author

Reviewer #1: This is my second review of the manuscript. I appreciate the authors’ adjustments to the text of the paper to more appropriately characterize their findings as non-causal. However, a number of my methodological concerns, outlined below, have not been addressed.

1. The justification for the use of zero-inflated negative binomial (ZINB) models remains unclear. The paper notes that there is over-dispersion because “the variance is much greater than the mean” (p. 6). But over-dispersion of count data involves comparing conditional variances to conditional means (not unconditional moments, as shown in Table 1). Is this the case?

2. The authors’ insistence on ZINB models also limits their ability to deploy district fixed-effects (FEs); as noted in the reviewer response document, ZINB models with district FEs fail to converge in their case. The use of these FEs is essential for rigorously accounting for unobserved spatial heterogeneity (including the concerns articulated around the localized outbreaks of acute encephalitis). The authors should consider the following Stata commands: reghdfe (for linear models with high-dimensional fixed effects, available at http://scorreia.com/software/reghdfe/) and ppmlhdfe (the corresponding command for Poisson models, available at http://scorreia.com/software/ppmlhdfe/). Both use efficient algorithms to absorb rather than estimate high-dimensional FEs and, in so doing, preserve degrees of freedom. At a minimum, report results from the linear model with the district FEs included. There is no need to report the coefficient for each estimated FE (indeed, neither reghdfe nor ppmlhdfe generate report these coefficients by default).

3. The authors’ reluctance to use the DLHS–3 in combination with DLHS–4 to generate a PHC-level panel is puzzling. The differences in how data are collected across the two waves do not rule out the possibility of pooling them. For instance, if DLHS–4 contains the total number and type of staff at the PHC while DLHS–3 only includes a binary variable for whether a particular type of staff member is present, a corresponding binary variable can be created in DHLS–4. The resulting panel admittedly contains less when it comes to that particular control variable but considerably richer as it enables controls for time trends and, perhaps most importantly, the use of PHC FEs if the same PHC is tracked in both survey rounds and there is temporal variation in access to electricity, i.e., access changes in some PHCs. Indeed, PHC FEs will fully account for the omitted variable bias that is almost certainly present in these analyses due to time-invariant characteristics, and the authors’ claim in the response that “this will still not help us account for potential endogeneity due to omitted variables” is incorrect. The tradeoffs between the panel and cross-sectional analyses can be articulated in the text; the authors don't need to pick only one approach.

4. My note in my original review that the concept of endogeneity is not limited to reverse causality concerns was intended to specifically highlight that the authors cannot rule out reverse causality. Consider, for instance, the following scenario: regional variation in population growth leads to an increase in PHC utilization (e.g., more deliveries) in some PHCs and not others. The government observes these trends and, in order to respond to this demand, makes investments in infrastructure at these PHCs, including electricity, i.e., utilization  electricity.

Reviewer #2: The authors have done a thorough job in revising the manuscript to address the comments made in the previous review round and have provided a clear and complete documentation of the revisions made. The revised manuscript makes the important supplementary information/analyses much more transparent, which adds much value to this manuscript. I have no further major revisions to suggest and recommend a couple of minor revisions as I outline below. I congratulate them for their good job.

1. In page 4-5, when authors say “To put this in context, as of 2019 there were still 795 PHCs without electricity supply. These PHCs without electricity supply cumulatively served at least 24 million rural individuals.”, it would be important to mention as to what does this mean in terms of percentages. For instance, 24 million rural individuals represent what proportion of India’s overall population, and similar 795 represent what proportion of unelectrified PHCs. This will help in indentifying the significance of the issue.

2. In response to my first comment from the first round, authors respond “We also note that between 2012 to 2019, the number of unelectrified PHCs reduced by 41% from 1919 to 795.” It would be important to highlight what does this progress mean in terms of percentages. Depending on the data availability, I would suggest them to add a time series line graph (somewhere in the annexure) that shows the progress in PHC electrification in India in the past 12-13 years (decline in percentage of unelectrified PHCs on Y-axis and year on the X-axis). They will use DLHS-3 proportion for the year 2007-08 and DLHS-4 proportion for 2012-13. For the remaining years, they can refer to yearly rural health statistics from MoHFW’s website.

7. PLOS authors have the option to publish the peer review history of their article (what does this mean?). If published, this will include your full peer review and any attached files.

Reviewer #1: No

Reviewer #2: **Yes: **Sunil Mani

---

## [Author Response · Author response to Decision Letter 1]

18 May 2021

Dear Dr. Zerriffi,

We take this opportunity to thank you and the two reviewers again for their positive feedback and further suggestions. We have implemented and reported both additional analysis that reviewer #1 suggested, regarding district fixed effects (point 2) and pooling data from DLHS-3 (point 3). As we have detailed in our attached response to reviewer document, you will see these additions reflected in new sections 3.4.4 and 3.4.5 (pages 21-23) in the revised manuscript. We find that our main results are robust to both these additional tests. Therefore, while the additional analyses do not change the overall conclusions of the study, they certainly increase our confidence in the results. 

Both rounds of the review process have been helpful in strengthening the findings and we appreciate the constructive suggestions from the reviewers. We believe we have responded sincerely to all outstanding concerns flagged by the reviewers. 

Regards,

Vivek

---

## [Editor Report · Decision Letter 2]

21 May 2021

Reduced health services at under-electrified primary healthcare facilities: Evidence from India

PONE-D-20-28579R2

Dear Dr. Shastry,

We’re pleased to inform you that your manuscript has been judged scientifically suitable for publication and will be formally accepted for publication once it meets all outstanding technical requirements.

Kind regards,

Hisham Zerriffi

Academic Editor

PLOS ONE

---

## [Editor Report · Acceptance letter]

26 May 2021

PONE-D-20-28579R2 

Reduced health services at under-electrified primary healthcare facilities: Evidence from India 

Dear Dr. Shastry:

I'm pleased to inform you that your manuscript has been deemed suitable for publication in PLOS ONE. Congratulations! Your manuscript is now with our production department. 

Kind regards, 

on behalf of

Dr. Hisham Zerriffi 

Academic Editor

PLOS ONE